# Turning Religious Experience into Reality: The Spiritual Power of Himma

Ismail Lala 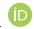

Department of Humanities and Social Sciences, Gulf University for Science and Technology, Mubarak Al-Abdullah 32093, Kuwait; lala.i@gust.edu.kw

**Abstract:** The extremely influential mystic, Muḥyī al-Dīn ibn 'Arabī (d. 634/1240), believes that the most advanced gnostics are imbued with a special power that turns their religious experience into reality. This is the power of *himma*—the power of existentiation that elite gnostics derive from God's absolute power of existentiation. Ibn 'Arabī and his followers assert that this power, which is exercised by the gnostics through an intense and unremitting concentration, actually shapes and forms external phenomenal reality as long as the concentration of the gnostic persists. This paper explores the different types of *himmas* that can exist, what kind of reality they allow the gnostics to perceive, and what relationship the objects created by *himma* have with the gnostic who exercised this power.

**Keywords:** religious experience; spiritual experience; mysticism; *himma*; Ibn 'Arabī; reality

## 1. Introduction

Religious experience has long been thought to inform our subjective reality. Friedrich Schleiermacher underscores the subjectivity of the experience, negating not only its phenomenological counterpart in others but also its articulation:

> But religion is of such a sort and is so rare, that whoever utters anything of it, must necessarily have had it, for nowhere could he have heard it. Of all that I praise, all that I feel to be the true work of religion, you would find little even in the sacred books. To the man who has not himself experienced it, it would only be an annoyance and a folly (Schleiermacher n.d., p. 14).

For Schleiermacher, religious experiences are as unique and manifold as those who experience them.[1] The sensitivity of such experiences, their disparate manifestations, and their myriad conduits, make religion in its entirety an endless panoply of phenomenological compilations, or as Schleiermacher puts it,

> as long as we are individuals, every man has greater receptiveness for some religious experiences and feelings than for others. In this way everything is different. Manifestly then, no single relation can accord to every feeling its due. It requires the sum of them. Hence, the whole of religion can be present only, when all those different views of every relation are actually given. This is not possible, except in an endless number of different forms (Schleiermacher n.d., p. 147).

Thus, religion is a sum total of all the subjective experiences of humankind. However, if this is the case, organised religion would be an ineluctable casualty since phenomenological subjectivity is afforded supreme authority.[2] Accordingly, there quickly appeared an appeal to objectivise these phenomenological subjectivities in order to preserve organised religion, and to affect an evolution in it. This tradition goes back to Philo (d. 50 CE), who believed that, along with the religious texts of Judaism, there were other non-textual sources that needed to be considered, as Harry Wolfson elucidates when he writes, 'Besides the written Scripture, Philo also draws upon certain unwritten traditions. These traditions are referred

to by him in various terms' (Wolfson 1962, p. 90). One of these terms was the 'unwritten law' (Wolfson 1962, p. 188), which Philo believed was a

> progressive revelation, a continuous revealment of God to chosen individual human beings to make known to them the meaning of the revealed Law. For though he believed that the revelation was final and perfect, inasmuch as the Law was to be eternal, this belief did not mean to him that it was a closed revelation (Wolfson 1960, p. 104).

Since these experiences were afforded the same epistemological value as scripture—the former being revelation of the gnostic, the latter revelation of a prophet—revelation was an ongoing divine act that was continually played out, but only in the overarching framework provided by sacred texts. It is God 'who reveals Himself in the depths of the self' to the gnostic in order to repristinate the religion (Scholem 1995, p. 34). Scholem describes this phenomenon in the following way:

> Revelation, for instance, is to the mystic not only a definite historical occurrence which, at a given moment in history, puts an end to any further direct relation between mankind and God. With no thought of denying Revelation as a fact of history, the mystic still conceives the source of religious knowledge and experience which bursts forth from his own heart as being of equal importance for the conception of religious truth. In other words, instead of the one act of Revelation, there is a constant repetition of this act. This new Revelation, to himself or to his spiritual master, the mystic tries to link up with the sacred texts of the old; hence the new interpretation given to the canonical texts and sacred books of the great religions (Scholem 1995, p. 28).

Therefore, the divinely revealed exegesis of scripture was perpetually being cast down in the world through the religious experiences of the gnostics. Muḥyī al-Dīn ibn ʿArabī (d. 638/1240), known as 'the Greatest Master' (*al-Shaykh al-Akbar*), due to the sheer enormity of the influence he exerted (and continues to exert) on Islam generally (Sufism specifically and our perception of the Qurʾan especially), agrees with this.[3] Indeed, he claimed that his principal works—*Fuṣūṣ al-ḥikam* and *Al-Futūḥāt al-makkiyya*—were products of mystical experience (Ibn ʿArabī 2002, p. 47; Schimmel [1975] 1978, p. 265). Such was the epistemological primacy of mystical experience that even the followers of Ibn ʿArabī predicated their religious insights on it (Morrissey 2020b; Lala 2019). This means that we can speak of an initial phase in which subjective experience informed the religious reality for the individual, which was then followed by a secondary phase in which this experience actually revived religious texts, thereby objectivising the subjective experience and making its application possible to those other than the individual who experienced it. This tethered the experience more closely with phenomenal reality. Nevertheless, Ibn ʿArabī did not just stop there, he believed that the experience of the gnostics of the highest level could actually *create* a phenomenal reality through the spiritual power of *himma*.

Ibn ʿArabī maintains all three of these levels, while at the same time acknowledging that the layfolk experience the religion subjectively but lack the authority to universalise their subjective experience as do the gnostics. It is then the gnostics of the highest level who can not only universalise their subjective experience in the interpretation of sacred texts but actually universalise it by creating phenomenal reality through the spiritual power of *himma*. This means a universalisation of their subjective experience is absolute since it is not restricted to those who believe in and read scripture but applies to all who experience sensible reality.

## 2. The Levels of Humankind

In the mystical *weltanschauung* of Ibn ʿArabī, humankind is afforded the highest rank since it possesses the potentiality to manifest all of God's 'most beautiful Names' (*Al-Asmāʾ al-ḥusnā*), such as 'the Compassionate', 'the Mighty', 'the Wise', etc., in one locus of divine

manifestation. These are the ninety-nine attributive Names of God that are mentioned in the Qur'an.[4] In the now extremely well-known beginning of the *Fuṣūṣ*, Ibn 'Arabī writes,

> God, be He praised, wanted, through His most beautiful Names that are countless, to see their essences. Or if you want, you could say, He wanted to see His essence in a comprehensive being (*kawn jāmi'*) that would comprise the whole matter because it has sensible existence, and with it, His secret would be manifest to Himself (Ibn 'Arabī 2002, p. 48).

Humankind has the ability to manifest all the divine Names, which is the purpose for the creation of the universe, as Ibn 'Arabī elucidates in this passage. Nevertheless, the capability to manifest the divine Names is one that is not realised by most people. Ibn 'Arabī explains this concept by means of an example:

> Our saying, 'Zayd is less knowledgeable than 'Amr' does not gainsay the essence of God being in the essence of Zayd and of 'Amr, but it is more perfect (*akmal*) and more knowledgeable in 'Amr than it is in Zayd, just as the divine Names are of different ranks but they are all still God (Ibn 'Arabī 2002, p. 153).[5]

The essence of the knowable God—which is God as He is described through the most beautiful Names (Lala 2021)—is manifested by 'Amr and Zayd. But the fact that 'Amr is more knowledgeable than Zayd means that he manifests God's Name 'the Knowing' (*Al-'Alīm*) more perfectly than Zayd (Al-Nābulusī 2008, 2:158–59; Al-Jāmī 2009, p. 367). This means that, even though the essence of humankind is one, inasmuch as they are all human, the potentiality to manifest the divine Names, and thus the epistemological rank that comes with it, differs. It also means that each person experiences reality, all of which is a manifestation of God's divine Names, in different ways. The susceptibility to experiencing reality as it truly is—a manifestation of God's Names in phenomenal reality—thus changes as the potentiality to realise the divine Names change.

Ibn 'Arabī affirms the value of subjective empiricism when he claims that even God 'gains' knowledge through His experience of the way in which His divine Names are manifested in the world. This is despite the fact that He has absolute knowledge of everything. If this 'dependence'[6] on phenomenological epistemology applies even to God, then it applies a fortiori to humans. He writes the following:

> He [Luqmān] describes God as 'Experienced' (*khabīr*), that is, knowing from experience (*ikhtibār*), just as God declared [in the Qur'an], '*And We shall surely try you until We know*' (Qur'an, 47:31). So this is knowledge from [spiritual] 'tasting' (*adhwāq*). Therefore, God makes Himself someone who gains knowledge, even though He knows the matter as it is. And we cannot deny what God clearly declares in the Qur'an regarding the truth about Himself. God, be He praised, therefore, made a distinction between the knowledge of [spiritual] tasting ('*ilm al-dhawq*) and absolute knowledge (*al-'ilm al-muṭlaq*) (Ibn 'Arabī 2002, p. 189).

Q47:31 asseverates that God's 'experience' of the universe is the basis of His empirical epistemology, which does not contradict His absolute knowledge of all things. Ibn 'Arabī seems to be saying that, even if one knows something, experiencing it and knowing it through 'spiritual tasting' is quite different. It is for this reason, adds Ibn 'Arabī, that God discloses that there are some of His servants who achieve such closeness to Him that 'I am his hearing through which he hears, his sight through which he sees, his hand with which he grasps, and his foot with which he walks' (Bukhārī 1987, 8:105).

The primacy of subjective experience in Ibn 'Arabī's mystical outlook is undeniable. Yet most people, as they do not actualise their potentiality to manifest the divine Names and do not achieve the level elucidated in the tradition above, must adhere to the letter of the law and cannot objectivise their experience by using it to interpret scripture. The gnostics, who have actualised their theophanic potentiality, however, are able to do this; as Ibn 'Arabī clarifies in the following:

> It is known that when the divine tongues of religions (*alsinat al-sharāʾiʿ al-ilāhiyya*) say about God, the Exalted, what they say, they do so in a way that conveys the immediate meaning to lay people (*al-ʿumūm*). As for the gnostics, they understand each word in many ways, no matter what language it is conveyed in. Therefore, God is manifested (*ẓāhir*) in every knowable thing while He is concealed (*bāṭin*) from all comprehension, except for he who says that the cosmos is His form and His essence (Ibn ʿArabī 2002, p. 68).

Ibn ʿArabī offers this as an explanation for Q57:3, which describes God as '*the First, the Last, the Manifest, the Hidden*'. The point he makes is that God, the Manifest, is experienced by lay people (*al-ʿumūm*) and gnostics alike. Now, even though all of the layfolk, as they are disparate theophanic loci, experience God, the Manifest, differently their experience does not qualify them to undertake the 'concealed' or inner exegesis of the Qurʾan, which itself is a scriptural representation of the universe; or as Ibn ʿArabī says, 'all [sensible] existence is letters (*ḥurūf*), words (*kalimāt*), chapters (*suwar*), and verses (*āyāt*), and that is the macrocosmic *qurʾān* (*Al-Qurʾān al-kabīr*)' (Ibn ʿArabī n.d.a; Lala forthcoming). This macrocosmic *qurʾān* is 'brought together' in the scriptural Qurʾan, which is why, according to Ibn ʿArabī, it is derived from the root *q-r-n* (to join together) and not *q-r-ʿ* (to read), as people generally assume (Ibn Manẓūr n.d., p. 3607; Lala forthcoming).

Lay people must therefore confine themselves to the 'immediate meaning', which is the 'basic meaning' of scripture (Izutsu 1998, pp. 18–24). Their subjective experience of God as the divine Names, of reality as it is the macrocosmic Qurʾan, and of the textual Qurʾan does not imbue them with the authority to excavate the 'concealed' (*bāṭin*) meanings of the Qurʾan since they have not fully actualised their potentiality to manifest the divine Names. This is not the case for the gnostics. They too experience God, reality, and the Qurʾan, subjectively. However, because they have fully realised their capability to manifest the divine Names, they have the authority to objectivise their subjective experience and employ them in the excavation of the 'concealed' meanings of the Qurʾan and the true reality of God and the universe about which it speaks. This is because, in their elevated experiences, they comprehend the true nature of God, of reality, and of the Qurʾan. This is what it means to 'understand each word in many ways'. Ibn ʿArabī highlights that the comprehension of the elite mystics is not limited to linguistic meanings of the Qurʾan nor even is it confined to language itself, which is why he makes it clear that the gnostics comprehend the true reality of the Qurʾan and, by extension, of God 'no matter what language it is conveyed in'.

The relationship between the divine Names, their manifestation as the universe, and of the epitomisation of the macrocosmic Qurʾan that is the universe as the scriptural Qurʾan, as well as the creational culmination of the divine Names as the gnostic who is the Perfect Man (*Al-Insān al-kāmil*)[7], may be summarised as the following (Figure 1):

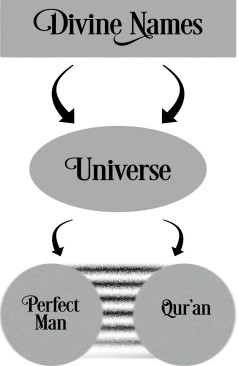

**Figure 1.** The divine Names as the Universe, the Perfect Man, and the Qurʾan.

This diagram makes it clear that there is complementarity between the gnostic, who is the Perfect Man, and the Qurʾan. Only when humankind fulfills its highest potentiality to manifest all the divine Names does its phenomenal experience have a direct analogue to

the reality of scripture. The subjective experience of reality, which is the divine Names writ large, can only be authoritative and amenable to universalisation, when all of the divine Names are faithfully manifested in the gnostic in a microcosmic way. Ibn ʿArabī, then, is in lockstep with Philo that the gnostics are able to use their own experiences as a vehicle to reinterpret and repristinate scripture. However, Ibn ʿArabī goes further. Since the universe is just a large-scale representation of the Perfect Man, the Perfect Man can—through his experience of reality and his unremitting concentration—project his experiential reality onto phenomenal reality. This is the spiritual power of existentiation known as *himma*, which is exercised by the very highest level of gnostics.

### 3. The Power of *himma*

*Himma* is not a term that readily lends itself to translation. As with almost all concepts and ideas in Ibn ʿArabī's worldview, the term has a Qurʾanic basis. In the chapter of Joseph, we are told that the wife of the man in whose house he resided pursued him. The Qurʾan says, '*Surely, she desired him (hammat bihī), and he would have desired her (hamma bihā)*' (Qurʾan, 12:24). The well-known Muʿtazilite exegete, Abuʾl-Qāsim al-Zamakhsharī (d. 538/1144), who is known for his linguistic exegesis of the Qurʾan (Ibrahim and Ibrahim 1980), explains that to '*hamma* a matter is to intend it (*qaṣadahū*) and to resolve on it (*ʿazam ʿayah*)' (Al-Zamakhsharī 1987, 2:455). Even though Edward Lane provides these denotations in his lexicon, he also writes that *hamma* lies in between intention and firm resolution (Lane 2003, 8:3044). Al-Zamakhsharī continues that to *hamma* a thing means 'to carry it out, and not to desist from it' (Al-Zamakhsharī 1987, 2:455). This means that, for al-Zamakhsharī, it is more than just an intention, it is a solid and unflinching resolution. This seems to be consistent with the way in which the term appears in Sufi lexicons.

Aḥmad al-Naqshbandī al-Khālidī (d. 1284/1867), the Ḥanafī mystic, in his lexicon of Sufi terminology, explains that there are three ranks of *himma*. He begins with 'the *himma* of awakening' (*himmat al-ifāqa*), writing, 'It is the first rank of *himma*, and it is the awakening of [the desire] to seek the One Who Remains (*Al-Bāqī*), and not seeking that which is ephemeral (*fānī*)' (Khālidī 1997, p. 97). The first level of *himma*, therefore, is the initial spiritual awakening of the gnostics after which they no longer seek the transitory world; they only seek God who will always remain. Al-Khālidī uses the divine Name 'The One Who remains' (*Al-Bāqī*) and juxtaposes it with the evanescence of the phenomenal world, in much the same way as the Qurʾan when it declares, '*All who are on it [the earth] will perish. And Your Lord's countenance will remain, radiant with majesty (jalāl) and honour (ikrām)*' (Qurʾan, 55:26–27). Subsequent to this, comes 'the *himma* of rejection' (*himmat al-anafa*), about which al-Khālidī remarks the following:

> It is the second rank, which inculcates in the one who possesses it rejection for seeking reward (*ajar*) for actions, to the extent that their heart rejects being occupied with expecting what God has promised in terms of reward for actions. It [the heart] then no longer fears witnessing God; rather, it worships God with perfection so that it does not fear turning to God, seeking only closeness to Him and nothing besides Him (Khālidī 1997, p. 97).

In this level, then, the gnostics do not seek the reward for worshipping God because they have risen above such mercenary pursuits and seek only proximity to the divine; their hearts, therefore, shun anything that is not God. However, even in this rank, there is a barrier between the gnostics and God because they do not seek God as He is in His absolute essence; they seek only the manifestation of God through the divine Names. This barrier to the divine in His absoluteness is removed in the final rank, which al-Khālidī says is, 'the *himma* of the lofty *himma* lords' (*himmat arbāb al-himam al-ʿāliya*). He writes the following:

> It is the third rank, and it is only concerned with God, and does not turn away from Him, for it is the highest of the *himmas*, inasmuch as [the gnostic] is no longer content with spiritual states or stations,[8] nor even, with stopping at cognisance of the divine Names and attributes, and desires nothing but the very Essence (*dhāt*) [of God] (Khālidī 1997, p. 97).

This means that the *himma* of the gnostic progresses from the initial spiritual awakening—as the desire to seek God—in the first stage, to rejecting anything that is not God—as a manifestation of the divine Names—in the second stage, to desiring only the absolute divine Essence in the final stage.

The Turkish mystic, Ḍiyā' al-Dīn Aḥmad ibn Muṣṭafā al-Kumushkhānawī (d. 1311/1894), provides a far more detailed analysis of *himma*, delineating nine aspects of it from its beginning to its end. His general definition of the term is as follows:

> Turning one's attention to God completely whilst rejecting all other considerations (*mubālāt*) by safeguarding the self from all other objectives (*aghrāḍ*) and reflections, and by adopting all the resources and means to achieve it, like [pious] actions, hope, and firm belief (*wuthūq*) in Him (Al-Kumushkhānawī 1913, p. 207).

The general contours of this definition correlate with al-Khālidī's assertions. However, al-Kumushkhānawī elaborates that this takes different forms and has different facets, based on the level of the gnostic. The form *himma* takes in the beginning stages (*bidāyāt*) is that '*himma* is attached to the obedience of God (*ṭā'a*)' (Al-Kumushkhānawī 1913, p. 207). This seems to be prior to the spiritual awakening with which al-Khālidī begins and paves the way for the spiritual awakening; for al-Kumushkhānawī affirms that only after this stage does the gnostic come to the gateways (*abwāb*) of *himma*, which is 'attachment of his heart to the felicity that always remains (*al-na'īm al-bāqī*), and his turning away from that which is ephemeral (*fānī*), and his diligence in seeking it without tiring (*tawān*)' (Al-Kumushkhānawī 1913, p. 207). Although al-Kumushkhānawī sets up the same juxtaposition as al-Khālidī by asserting that the gnostic seeks that which remains (*bāqī*) and scorns that which is ephemeral (*fānī*), i.e., the world, yet there is a fundamental difference between them. This is because al-Khālidī believes that what is sought of the things that remain is God, whereas al-Kumushkhānawī declares it is paradise.

Al-Kumushkhānawī continues that the conduct of those with *himma* towards others (*mu'āmalāt*) is that that their *himma* 'evokes them to remain steadfast in [righteous] deeds whilst remaining vigilant of the [lower] self (*murāqaba*) and having firm trust in, and completely submitting to, God (*al-tawakkul wa'l-taslīm*)' (Al-Kumushkhānawī 1913, p. 207). This means that all their social interactions are guided by these principles of righteousness, self-evaluation, and trust in God, and they pay no attention to how things ostensibly appear. This, then, informs all their actions, and their disposition (*akhlāq*) becomes one in which they 'turn their *himma* completely towards the acquisition (*iḥrāz*) of felicity and perfections (*kamālāt*)' (Al-Kumushkhānawī 1913, p. 207). So far, however, the gnostic still seeks felicity and not the divine itself, which is the initial stage al-Khālidī mentions. But when it comes to the principles (*uṣūl*) of *himma*, al-Kumushkhānawī explains that 'it attracts its possessor to the divine presence, with the strength of certitude (*yaqīn*), and the soul of intimacy (*rūḥ al-uns*), which prevents lassitude (*futūr*) in proceeding on the spiritual path and from deviating (*zaygh*) from one's purpose' (Al-Kumushkhānawī 1913, p. 207). This corresponds with al-Khālidī's first rank because the purpose is now the divine presence and nothing else.

Al-Kumushkhānawī then expatiates on the 'states' (*aḥwāl*) when all the disparate '*himmas* become one *himma* seizing (*istīlā'*) divine love' (Al-Kumushkhānawī 1913, p. 207). The gnostic has now dissociated from the world and is cultivating their relationship with God alone. Next come the ranks of sainthood (*wilāyāt*) when the *himma* 'rises from the states and stations (*maqāmāt*) to the plane of the divine Names and attributes' (Al-Kumushkhānawī 1913, p. 207). This is the second rank that al-Khālidī describes when the gnostics seek only God just as they conceive of Him in terms of His Names and attributes. This changes in the following level of 'the realities' (*al-ḥaqā'iq*) when 'the *himma* rises above the divine attributes and turns away from the characteristics of God to the divine Essence (*dhāt*) (Al-Kumushkhānawī 1913, p. 207). Now the final rank al-Khālidī delineates has been attained wherein it is the absolute divine Essence that is sought and not God as He relates to His creation through His divine Names and attributes. But al-Kumushkhānawī writes that,

even after this, there is a rank, which he names 'the final stages' (*al-nihāyāt*). He writes that, in this stage

> there is no *himma* except perception through effects of God in all existent things, like when God said, '*And you [Muḥammad] did not throw what you threw, but it was God Who threw [it]*',[9] and when He said, '*And when you brought forth the dead with My permission*'.[10] So in this final stage, ostensible actions and earning rewards (*takassub*) are annihilated and it is unsullied by the contamination of contingency; the path becomes widened and expansive, and the heart ascends to the station of the [divine] secret (*maqām al-sirr*) (Al-Kumushkhānawī 1913, p. 207).

The final stage, as is clear, is merely a progression from the rank of the realities and adds further certainty to the heart of the gnostic that all things in creation are ultimately manifestations of the divine Essence. The gnostic, thus, becomes aware that there is no real contingency, only the absolute existence of the divine manifested as contingent beings. Al-Kumushkhānawī, therefore, agrees with the basic tripartite levels that al-Khālidī delineates, but he adds further sublevels to these in his analysis. The reason for this concordance is detailed below.

Ibn 'Arabī's categorisation of *himma* in his *Rasā'il* agrees with these Sufi manuals since he defines it as 'isolating the heart for the objects of desire (*munā*)' (Ibn 'Arabī 1997, p. 536), but this is a characteristically ambiguous definition. Further detail provided by Henri Corbin, elucidates that *himma* is actually very different for Ibn 'Arabī because he believes it is

> the act of meditating, conceiving, imagining, projecting, ardently desiring . . . It is the force of an intention so powerful as to project and realize a being external to the being who conceives the intention (Corbin 1997, p. 222).

When elite gnostics focus their absolute and unremitting intention to the extistentiation of a being in the phenomenal world, they cause them to exist in it. Su'ād al-Ḥakīm explicates that, according to Ibn 'Arabī, *himma* is 'an active faculty' (*quwwa f "āla*) or an 'active capability' that human beings have (Ḥakīm 1981, p. 1109). This faculty or capability may be given by God and be part of the 'natural constitution' (*jibilla*) of a person, or it could be the fruit of 'nurture' (*tarbiya*) and 'acquired' (*iktisāb*) (Ḥakīm 1981, p. 1109). Ibn 'Arabī specifies the disparate provenances of *himma* in his work, *Mawāqi' al-nujūm*:

> Know that the existence of this *himma* in the servant is of two kinds . . . a *himma* that arises in the essence of the inborn disposition (*khilqa*) of a servant and [in his] natural constitution (*jibilla*), and a *himma* that is gained after not having it (Ibn 'Arabī 1907, p. 84).

As an example of the *himma* of natural constitution, Ibn 'Arabī mentions the incident of 'Īsā speaking to defend his mother's honour whilst he was still 'in the cradle' (*fi'l-mahd*), as detailed in Q19:29-33 (Ibn 'Arabī 1907, p. 84). The proof of the *himma* of acquisition, says Ibn 'Arabī, is in the Prophet Muḥammad's saying, 'Learn certainty (*yaqīn*)', and in the story of 'Īsā, because 'when it was said to him that he used to walk on water, [he replied that] if he had increased his certainty, he would have walked on air!' (Ibn 'Arabī n.d.b). Ibn 'Arabī explains that the level of certainty (*yaqīn*) in God, as the ultimate cause of all things in the phenomenal realm, directly impacts our perception of it and our capabilities within it such that miraculous feats, like walking on water or floating on air, are achieved thereby.

As *himma* may be innate or acquired, and it is an active capability that allows the manipulation of reality, it can be employed for diverse objectives, as al-Ḥakīm elaborates in the following:

> Since *himma* is only a capability, it varies according to that to which it is attached, and it follows the will (*irāda*) of its bearer. So if the bearer of *himma* attaches it to the [material] world, we see him gain worldly treasures . . . and if he attaches it to worship ('*ibāda*), he will attain [spiritual] stations (*maqāmāt*) . . . and if he

attaches it to God, all other attachments (*ta'alluqāt*) fall away and his multiple *himmas* become one *himma* (Ḥakīm 1981, p. 1111).

There is a reciprocity between the power of *himma*, which is active, and its object, which is passive, such that the passive object also determines the active power of *himma*, thereby becoming active and rendering the power of *himma* passive.

Ibn 'Arabī allows the power of *himma* to be attached to the material world and if this is the case the bearer of this *himma* gains what they desire. There are others who focus their power of *himma* on religious worship, which allows them to ascend the stations and progress on their spiritual journey. But even this, intimates Ibn 'Arabī, is a degree removed from a pure focus on God. This second group is subdivided by al-Kumushkhānawī into multiple initial groups that seek felicity in the hereafter through their *himma*. The third cohort, says Ibn 'Arabī, comprises those spiritual elites who merge their 'multiple *himmas*' into one *himma* with the sole objective of reaching God. Al-Kumushkhānawī also speaks of combining disparate *himmas* into one *himma* to seize divine love in the rank of 'states' (*aḥwāl*). This means that not only are there different kinds of *himma*—from those applied to the profane as well as to the sacred—but the same person can have multiple *himmas*, just as they have multiple inclinations and objects of desire. Ibn 'Arabī elucidates that the matter to which the power of *himma* is applied plays a pivotal role in the power itself:

> Surely the *himmas* vary according to the varying objects of desire (*maṭāmi'*) because the *himma* is attached to them . . . and, were it not for the objects of desire, the *himma* would be cut off, and if there was no *himma* then there would be no actions (*a'māl*) (Ibn 'Arabī n.d.b, p. 15).

As the basis for the action, then, *himma* is crucial for anything to be achieved, but al-Ḥakīm explains that 'powerful *himma*' (*al-himma al-qawiyya*) is more than just intention and desire, it is in the 'root of the natural constitution' (*jibilla*) and allows the bearer to 'ascend ranks' because it is 'attached to prodigious affairs' (*'aẓā'im al-umūr*) (Ḥakīm 1981, p. 1111).

Ibn 'Arabī, like al-Khālidī, declares that there are three grades of this *himma*. The first one he calls 'the *himma* of [mystical] awakening' (*tanabbuh*). He writes the following:

> The *himma* of [mystical] awakening (*tanabbuh*) is the heart's waking up to what the reality of humankind bestows, which is what one's desire is attached to, whether it is impossible or possible, so it is isolating the heart for the objects of desire (*munā*) (Ibn 'Arabī n.d.a).

This is the level Ibn 'Arabī begins with in his definition of *himma* in the *Rasā'il* (see above). It differs from al-Khālidī's first level of spiritual awakening when the world is already forsaken by the gnostic for the divine. Corbin explains that, for Ibn 'Arabī, this level of *himma* gives a person the capacity to become cognisant of things as they really are. Therefore, it provides information about the object of desire that cannot be gained by the intellect and is only attained through mystical 'tasting' (*dhawq*) (Corbin 2008, pp. 269–70). Ibn 'Arabī writes the following:

> This *himma* makes him [the bearer] truly 'perceive' that which he desires . . . so if this perception gives him [the inclination] to withdraw from the object of desire, then he withdraws; and if it gives him resolution to pursue it, then he becomes resolved (Ibn 'Arabī n.d.a).

Schleiermacher makes the same point about the 'sages' who perceive the reality of things residing, as they truly are, under their phenomenal manifestations. This is their 'clear intuition', in which

> all strife between appearance and reality is resolved, and who, therefore, undisturbed by these refinements, can again be stirred like children, their joy would be a real and pure feeling, a living impulse, a gladly communicative contact between them and the world (Schleiermacher n.d., p. 56).

These sages experience the world in a completely different way, and they are able to reconcile the dichotomy between the appearance of things in the ostensible universe and

their inner reality. The sage is only able to achieve this, Schleiermacher asserts, because 'his nature is reality which knows reality' (Schleiermacher n.d., p. 33). When he has reached this level, the gnostic becomes cognisant of the connection between him and the universe and so he is able to

> pursue the play of nature's powers into their most secret recesses, from the inaccessible storehouses of energized matter to the artistic workshops of the organic life. He measures its might from the bounds of world-filled space to the centre of his own Ego, and finds himself everywhere in eternal strife and in closest union. He is nature's centre and circumference. Delusion is gone and reality won (Schleiermacher n.d., p. 100).

For Ibn 'Arabī, the reason he finds 'he is nature's centre and circumference' is because he has now realised his inner reality as a locus of manifestation of the divine Names, and that the reality of the universe is identical to his, but on a macrocosmic scale. This is his mystical awakening according to Ibn 'Arabī. And since the *himma* of mystical awakening allows the gnostic to gain esoteric knowledge about the reality of the object of their desire, says Ibn 'Arabī, he is now in a position to make an informed decision about whether to pursue it or not. Based on this, the gnostic resolves to acquire the object or to abandon it. However, because the epistemological basis of the reality of the object is mystical, it is inaccessible to the intellect and can only be gained through 'tasting'. Ibn 'Arabī frequently refers to knowledge that he gained only through mystical tasting.[11] Schleiermacher makes the same point about the sages who gain this faculty (Schleiermacher n.d., pp. 45–46, 56, 100). Ibn 'Arabī writes in the *Fuṣūṣ* that this knowledge of tasting is predicated on the rank of the gnostic: 'the knowledge of the divine through [mystical] tasting (*al-'ulūm al-ilāhiyya al-dhawqiyya*) that the people of God, the Exalted, have varies depending on differing abilities' (Ibn 'Arabī 2002, p. 107). So, at this stage, the *himma* bequeaths the gnostics true perception of reality, which is what things truly are behind the phenomenal façade that everyone else views. This level of *himma* shapes their reality, but it does not affect the reality of others. The second level, however, affects others as well.

Ibn 'Arabī dubs the *himma* of the second level, 'the *himma* of the will' (*irāda*). He writes the following:

> As for the *himma* of the will . . . it is a comprehensive *himma* (*himma jam'iyya*) . . .
> so if the self (*nafs*) comes together, it can affect bodies of the [sensible] world
> (*ajrām al-'ālam*) and their states (*aḥwāl*) (Ibn 'Arabī n.d.a).

This is the power of existtentiation that is derived from the divine power of existentiation. It is a level that al-Khālidī or al-Kumushkhānawī do not mention. Ibn 'Arabī explains that, when the gnostics of the highest level use their spiritual power of *himma* that comes from their enlightened hearts, they are able to mirror the divine power of existentiation, and God uses them as 'causes (*asbāb*) [through which] God, be He praised and exalted, does things that are already [determined] by Him' (Ibn 'Arabī 1907, p. 84). The elite gnostics, thus, employ the power of *himma* only in accordance with the divine will and not based on their own desires (see below).

This power of existentiation emanates from the spiritual concentration of the elite gnostics, as long as it persists. Ibn 'Arabī clarifies that the elite gnostics do not actually 'create' anything in phenomenal reality, the operation that their power of *himma* carries out is the transfer of already existing beings in the pre-phenomenal realms of reality that are then manifested in the sensible world (Corbin 1997, pp. 225–26).[12] Their spiritual concentration maintains the perpetual existence of the object in the sensible world because Ibn 'Arabī agrees with Ash'arite occasionalism that our perception of phenomenal reality is rapid perpetual re-creation by God.[13] He, nevertheless, believes that the Ash'arites only got this partially right as they dismissed the ever-present divine substrate that underpins all of reality (Ibn 'Arabī 2002, p. 156).

Since all of reality is merely a rapid re-creation of things in different realms, and because all these realms are connected—being nothing but more and more differentiated

versions of the divine Names of God (Lala 2022)—the elite gnostic is able to affect the re-creation of things from the pre-phenomenal realms to the phenomenal one through their power of *himma*. The power of *himma* is therefore the power to cause the phenomenal appearance of things that are already present in other pre-phenomenal realms (Corbin 1997, p. 226). This power, then, creates a new phenomenal reality as long as their spiritual concentration persists, but it may only be other gnostics who can perceive this new reality (Corbin 1997, p. 227).

Ibn 'Arabī then mentions the third and highest level of *himma*. This is 'the *himma* of the Reality' (*al-Ḥaqīqa*), 'which is the combination of *himmas* with the purity of inspiration (*ilhām*)' (Ibn 'Arabī n.d.a). He says that this *himma* is reserved for the gnostics of the very highest level 'who combine their *himmas* on God' (Ibn 'Arabī n.d.a) because He is the only one with true reality.[14] These elite gnostics use this combination of *himmas* to connect with the 'divine unity' (*aḥadiyya*), which is a term Ibn 'Arabī customarily reserves for the numinous God in His undifferentiated state (Lala 2019, pp. 119–23, 164; Al-Qāshānī 1992, p. 51; Jurjānī 1845, p. 12). In this way, the gnostics of the highest level are able to see the unity behind phenomenal multiplicity, or, as Ibn 'Arabī puts it, they do this 'to flee from multiplicity, and seeking the oneness of multiplicity (*tawḥīd al-kathra*), or just oneness' (Ibn 'Arabī n.d.a). The highest level gnostics, then, do not just perceive things as they truly are like the gnostics of the first level. They are able to go beyond the reality of things and see them as a manifestation of the ultimate reality which is God. Al-Khālidī and al-Kumushkhānawī also afford this rank to the highest level gnostics. This means that the elite gnostics see the oneness in the multiplicity of creation, which in turn means their perception of reality is entirely distinct from everyone else. Al-Ḥakīm elaborates that, as the gnostics ascend through the ranks of spirituality, they lose their attachment to all things besides God until there remains no other object for the *himma* to be applied to except God; therefore, it is in this way that all the *himmas* become just one *himma* (Ḥakīm 1981, pp. 1111–12).

Ibn 'Arabī's conception of *himma*, although it bears similarities with al-Khālidī and al-Kumushkhānawī, is clearly different. One of the reasons for this is that 'Abd al-Razzāq al-Qāshānī (d. 736/1335?), arguably the most influential person for the formalistion of Ibn 'Arabī's thought, did not simply perform the function of systematising and promulgating Ibn 'Arabī's ideas, he also highlighted some concepts and downgraded others (Lala 2019). His treatment of *himma* in the most comprehensive of his three lexicons on the Sufi nomenclature of Ibn 'Arabī and the final work he authored in his life, *Laṭā'if al-i'lām*, bears ample testimony to this (Al-Qāsimī n.d., p. 733). In this work, al-Qāshānī begins by offering the same basic definition of *himma* as Ibn 'Arabī in the *Rasā'il*. He also agrees that *himma* primarily denotes attachment of the heart to God, not to the world or even to the reward God bestows on His faithful servants (Al-Qāshānī n.d., 2:335). It is for this reason, he asserts, that *himma* has been defined as 'seeking God, whilst shunning (*i'rāḍ*) all things besides Him, without lassitude (*futūr*) or tiring (*tawān*)' (Al-Qāshānī n.d., 2:335). Al-Qāshānī then adduces the same tripartite classification of *himma* into the initial '*himma* of awakening' (*himmat al-ifāqa*), then 'the *himma* of rejection' (*himmat al-anafa*), and, finally, 'the *himma* of the lofty *himma* lords' (*himmat arbāb al-himam al-'āliya*) as al-Khālidī (Al-Qāshānī n.d., 2:335–36), indicating that the latter enthusiastically adopted the former's definitions. This would add credence to the contention of recent scholarship that al-Qāshānī exerted a powerful and abiding influence on the intellectual topography of Sufism (Lala 2019).

Al-Qāshānī writes that the '*himma* of awakening' (*himmat al-ifāqa*) is when 'the heart of the servant [of God] awakens from the overwhelming effects of temporal transience (*duhūr*) and the tribulations of desires, so he sees the world in this state as repugnant'. This is because in this state, adds al-Qāshānī, 'he sees that everything in the world has no permanence, so he craves that which abides because he sees that the hereafter (*ākhira*) does not have an ending' (Al-Qāshānī n.d., 2:336). It is clear that, even though al-Khālidī accepts al-Qāshānī's terminology wholesale, his definition of the '*himma* of awakening' is slightly different from his predecessor because, whereas al-Qāshānī states that it is a spiritual awakening that galvanises the gnostic to seek the hereafter because it will remain

forever, unlike this ephemeral world, al-Khālidī affords the gnostic the higher rank of seeking God Himself.

Al-Qāshānī's second rank of 'the *himma* of rejection' (*himmat al-anafa*), wherein the gnostic does not seek rewards for their actions, but instead seeks only proximity to the divine (Al-Qāshānī n.d., 2:336), is fully adopted by al-Khālidī, who produces a strikingly similar account of it in his lexicon. Al-Khālidī's final rank of 'the *himma* of the lofty *himma* lords' (*himmat arbāb al-himam al-'āliya*), nevertheless, furnishes the reader with far more detail than al-Qāshānī divulges, for he writes simply that it is the rank in which only God is sought by the mystic, to the exclusion of all else, which is why this *himma* is called 'the highest of the *himmas* because it is related to God, above Whom there is nothing' (Al-Qāshānī n.d., 2:336). 'This is why', adds al-Qāshānī, 'his *himma* is also called "the lofty *himmas*"' (*al-himam al-'āliya*) (Al-Qāshānī n.d., 2:336). It is this rank that al-Khālidī describes as 'the *himma* of the lofty *himma* lords' in which the mystic's *himma* 'does not even stop at witnessing the divine attributes; rather, it goes beyond witnessing the divine qualities to the divine essence itself' (Al-Qāshānī n.d., 2:337).

It may be clearly discerned from the foregoing that al-Qāshānī heavily influenced al-Khālidī, who presents a condensed version of his entire treatment of *himma* with only slight changes. If al-Qāshānī exerted a powerful influence on al-Khālidī, then his impact on al-Kumushkhānawī is even greater as his entire disquisition on *himma* has been taken verbatim from al-Qāshānī's *Iṣṭilāḥāt al-ṣūfiyya* (Al-Qāshānī 1992, pp. 304–5), which is why al-Kumushkhānawī's analysis reads as just a more detailed version of al-Khālidī's. Al-Qāshānī's formalisation of Ibn 'Arabī's thought evidently achieved its intention of garnering a much wider audience (Lala 2019; Al-Qāshānī 1892, p. 3; 1992, p. 21; 1995, p. 34). However, in order to achieve this goal, al-Qāshānī either downplays aspects of Ibn 'Arabī's thought or omits them entirely (Lala 2019). Specifically, in this case, there is no mention of the power of existentiation that features so prominently in Ibn 'Arabī's works in either the *Iṣṭilāḥāt* or the *Laṭā'if*. Ibn 'Arabī spills a lot of ink in describing the '*himma* of the will' that enables the gnostics of the highest level to shape reality, but al-Qāshānī overlooks it in these two lexicons as it does not serve his purpose. He does, nevertheless, include it in his lexicon of intermediate length, *Rashḥ al-zulāl*, where he writes the following:

> This *himma* is called 'the *himma* of the will' (*himmat al-irāda*), and it is a comprehensive *himma* (*himma jam'iyya*); the self (*nafs*) becomes restricted to it so nothing can oppose it, to the point that if he [the gnostic] conceptualises something and wishes it to exist, it would exist immediately (Al-Qāshānī 1995, p. 116).

This definition of 'the *himma* of the will' is completely consistent—in meaning and nomenclature—with Ibn 'Arabī's articulation of it in the *Futūḥāt*. However, al-Qāshānī side-lines this concept, which he perhaps deems divisive, and omits it from his lexicon for the initiates, *Iṣṭilāḥāt al-ṣūfiyya*, and his most comprehensive and final work, the *Laṭā'if*. Whereas the idea that the perception of reality of the gnostics of the highest level differs from layfolk is comparatively uncontroversial—which is why al-Qāshānī dedicates so much effort to its delineation and formalisation in the *Iṣṭilāḥāt* and then the *Laṭā'if*—the belief that these gnostics could actually alter reality might prevent initiates from embarking on the mystical path or neophytes from proceeding along it. Al-Qāshānī's careful curation of Ibn 'Arabī's mystical outlook entails the relegation of the notion that the highest-level mystics change reality itself. But if this is the case, as Ibn 'Arabī argues and even al-Qāshānī concedes, and gnostics can affect phenomenal reality through their '*himma* of the will', why is it that we do not often find gnostics exercising this awesome power?

## 4. Using *himma* to Existentiate Sensible Reality

The power of *himma* can be used by gnostics to do miraculous things (*karāmāt*) in the phenomenal world, as Ibn 'Arabī elucidates, 'And they are not, I mean miracles [worked by gnostics], except that which is manifest through the power of *himma*' (Ḥakīm 1981, p. 1116). The remit of the power of *himma* to form reality is vast. In fact, Ibn 'Arabī writes that it encapsulates all things that can be done through ostensible means, 'Everything that

cannot be gained by a person except by their body or by apparent means (*sabab ẓāhir*), is gained by the prophet or the saint (*walī*) through their *himma*' (Ibn 'Arabī 1907, pp. 83–84). Yet the gnostics seldom use this power, as Samuela Pagani observes, 'Les saints et les prophètes dont la connaissance est plus parfaite, évitent cependant d'avoir recours à ces pouvoirs' (Pagani 2014, p. 121). Partially, this is to protect the neophytes who are not spiritually equipped to witness the incredible power to form a phenomenal reality that the elite gnostics wield; therefore, this power would only try their faith if they saw what the gnostics could do (Ibn 'Arabī n.d.b). But it is also because

> they knew that in this world the servant cannot become the Lord, and that the subject who dominates a thing (*mutaṣarrif*) and the thing he dominates (*mutaṣarrif fīhi*) are essentially one being, but also because they recognized that the form of what is epiphanized (*mutajallī*) is also the form of what the epiphany is revealed (*mutajallā-lahu*) (Corbin 1997, p. 229).

This is why, when Zakariyya did eventually use this power to pray for a son, while he was senescent and his wife barren (Q19:1-6), he invoked the divine Name, 'The Master' (*Al-Mālik*), as Ṣadr al-Dīn al-Qūnawī (d. 673/1274), the adopted son and foremost disciple of Ibn 'Arabī (Todd 2014), clarifies in the following:

> Know that the secret that describes his wisdom is the wisdom of acquisition because what overwhelmed his state was that it was governed by the [divine] Name, 'The Master' (*Al-Mālik*). This is because dominion (*mulk*) is power (*shidda*), and an owner (*malīk*) is powerful (*shadīd*). And God is *the possessor of power (dhu'l-quwwa), The Strong (Al-Matīn)* (Qur'an, 51:58). So God aided him with power that penetrated his *himma* and concentration, and produced a response and acquisition of what was asked for (Qūnawī 2013, p. 106).

It was Zakariyya's *himma*, then, that allowed him to access God's absolute lordship and His absolute power of existentiation, which, in turn, enabled him to create the reality that he wanted. However, not only was he reticent about using this power, he made his supplication privately so that the layfolk would not be overawed by his power, and in order to maintain his spiritual concentration that allowed existentiation of such a reality (Al-Jāmī 2005, p. 162).

Perhaps the best example of the lengths gnostics and prophets go to avoid using the power of *himma* is the prophet Lūṭ. In the Qur'anic narrative, Lūṭ, overwhelmed by the transgression and recalcitrance of his people, cries out, '*If only I had power to oppose you or recourse to some strong support*' (Qur'an, 11:80). Ibn 'Arabī believes that Lūṭ already had recourse to 'strong support' (Ibn 'Arabī 2002, p. 127); it was his power of *himma* through which he could have shaped reality to whatever he desired, but he chose not to use it. The Sufi gives two reasons for this: (1) 'because of his realisation of the station of servanthood (*maqām al-'ubūdiyya*)', and (2) because there is a 'unity of the one acting and the one acted upon' (Ibn 'Arabī 2002, p. 128). These are the same reasons Corbin delineated. The principal student of al-Qūnawī, Mu'ayyid al-Dīn al-Jandī (d. 700/1300?), who studied under him for ten years (Kaḥḥala 1993, p. 943; Baghdādī 1951; Al-Jandī 1982, p. 12, 2:484; Ḥājī Khalīfa 1941, 2:1261) and wrote a commentary on the *Fuṣūṣ* under instruction from al-Qūnawī (Al-Jāmī 1858, pp. 648–50), explains that this means there is a

> unity between the one acting and the one who is acted on, which demands acting even as it prevents acting because it occurs in the same thing; there being nothing in existence except God. But the [phenomenal world] requires actions to occur, so if the gnostic does act, his act is nothing but the act of God, be He praised, on account of the aforementioned unity. This is particularly true for the perfect servant (*al-'abd al-kāmil*), who has taken on all of the divine nominal realities (*al-ḥaqā'iq al-asmā'iyya al-ilāhiyya*) of the Lord so that none of his traits of servanthood (*ṣifāt al-'abdāniyya*) remain because of the unity of the essence, for if they do, he is not a perfect servant. Yet he does not exercise or wield his power of *himma* for fear that he forsakes the station of servanthood (Al-Jandī 1982, p. 402).

Since all things in reality are theophanic loci, the division between agent and object, active and passive, is effaced. Nevertheless, al-Jandī appreciates that the semblance of causality has to be maintained, and this may require the exercise of the power of *himma*; so when a gnostic—who is the Perfect Man or the perfect servant because he has actualised his potentiality to manifest all the divine Names, and thus 'taken on all of the divine nominal realities'—uses his power of *himma*, his action is nothing but God's action. By wielding this immense power to form reality, the gnostic is operationalising the divine power of existentiation, which means that now he is using his manifestation of the divine Names of lordship to exercise complete control over phenomenal reality. This is why 'none of his traits of servanthood (*ṣifāt al-'abdāniyya*) remain' in him. But the gnostic does not want to flaunt this active power; he wishes to remain in 'the station or servanthood' so he does his best to not use his power of *himma* unless it is absolutely necessary, al-Jandī puts it as follows:

> The only thing in reality that stops him [from using his *himma*] is [wanting] to stay in the station of essential servanthood (*maqām al-'ubūdiyya al-dhātiyya*), which is [suitable] for him, and giving the trust of accidental lordship (*al-rubūbiyya al-'araḍiyya*) back to God, following the example of the people close to God. Therefore, he does not busy himself with acting [effectively] or controlling [others], instead turning his attention entirely to God (Al-Jandī 1982, p. 402).

The gnostic refrains from using his incredible existentiating power of *himma* and stays in the station of servanthood by giving back to God the power to exercise lordship over the sensible world. This power of lordship that the gnostic returns is accidental lordship that is derived from God's absolute lordship. Therefore, he remains a passive servant and does not become an active lord in the world. The gnostics of the highest level are the ones least likely to use their power of *himma* because they see reality as it truly is—a manifestation of the divine Names—so they have no desire to change it, as al-Jandī makes clear in the following:

> Perfection in gnosis and knowledge about the realities of things necessitates perfect comportment before God, the Exalted, which is not occupying oneself with manifesting power and influence through the spiritual power of *himma* (Al-Jandī 1982, p. 405).

It is for this reason that Lūṭ, despite having the spiritual power to manipulate sensible reality, did not use it even when he was overwhelmed by his people. More generally, this is why gnostics do not ordinarily exercise their power of *himma* unless it is in accordance with the divine will. The power of existentiation that the gnostics have, thus, is just another one of the causes God employs to carry out His decree in the phenomenal world; it is never employed by the gnostics because of their desires since they have achieved the rank of the Perfect Man. This means that they have become a locus of manifestation of all the divine Names. Due to being a comprehensive theophanic locus, there is no separation between the gnostic and the divine Names, and so God's will is the gnostic's will. The power of *himma*, then, is deployed is pursuance of this singular will.

## 5. Conclusions

It has long been acknowledged that religious experiences inform our subjective reality, but this could be antithetical to the dictates of organised religion if it was given absolute priority in every case and for every person. Thus, there was a move to objectivise these phenomenological subjectivities by affording only elite gnostics a parallel revelation in addition to the initial revelation received by prophets. This perpetual revelation not only allowed a repristination of the religion in accordance with unique concerns of the gnostics from different eras, but it also maintained a direct channel between God and His closest servants. Ibn 'Arabī took this process of objectivisation a step further by claiming that the gnostics of the highest level do not just perceive reality in a different way than normal people because they view the true divine reality behind the phenomenal façade, they can

actually form phenomenal reality through their mystical power of *himma*, which enables them to transfer existents from pre-phenomenal realms to our physical world. This means there is a progression from the subjective reality of lay people to objective scriptural reality, based on the subjective reality of gnostics, to objective phenomenal reality, based on the subjective reality of the highest level gnostics. However, Ibn ʿArabī makes it clear that the gnostics of the highest level never use this power in pursuance of personal desires. In their capacity as comprehensive loci of all the divine Names, or Perfect Men, they are in complete accord with the divine Will and thus their power of *himma* becomes another conduit for the execution of the divine Will in phenomenal reality.

**Funding:** This research received no external funding.

**Acknowledgments:** I would like to express my gratitude to the anonymous reviewers of this work who, through their insightful comments, allowed me to clarify many aspects that were left undefined in the previous draft. I would also like to thank Noor Almujeem for her assistance in the creation of the diagram.

**Conflicts of Interest:** The author declares no conflict of interest.

## Notes

1　One must differentiate between religious experiences and spiritual experiences. Whilst the former is associated with an articulation of experiences that are within a specific religious context, the latter is more often used to denote a private experience (Hood 2009, p. 189). It is clear that Schleiermacher conflates these terms and blurs the lines between the private experience and the religious expression of it.

2　The issue of subjectivity and religious diversity is discussed by (Alston 2014), chapter 7. Gershom Scholem argues that it is evidently untenable, in the context of experiences that emanate from and are operative within specific religious beliefs, to speak of experiences that are purely abstract. He writes, 'There is no such thing as mysticism in the abstract, that is to say, a phenomenon or experience which has no particular relation to other religious phenomena. There is no mysticism as such, there is only the mysticism of a particular religious system, Christian, Islamic, Jewish mysticism and so on' (Scholem 1995, p. 26). One must therefore make a distinction between mystical experiences and religious experiences. But even before this can be done, one should attempt to define mysticism. As William Ralph Inge observes, however, there is no universally agreed-upon definition of mysticism, which is why he provides approximately twenty-six definitions of the term (Inge 2010). Scholem agrees with this assessment (Scholem 1995, pp. 23–32), but that is not to say that one cannot make any pronouncements about mysticism and mystical experience. Mystical experience—as it is most commonly conceived—forms a subcategory of religious experience. As such, a mystical experience is a kind of religious experience (Webb 2022). This being the case, all religious experiences would be mystical experiences, but not vice versa. For the purposes of this paper, in the context of Ibn ʿArabī's works, I shall be referring to religious experiences that are articulations of spiritual experiences, and narrowly conceived as mystical experiences, in order to mitigate issues arising from differing definitions.

3　A detailed exploration of the life of Ibn ʿArabī and his thought, in general, is outside the scope of this study. For the former, see (Addas 1993; Hirtenstein 1999). For the latter, there is a wealth of material available, such as (Izutsu 1983; Landau 2008; Sells 1994; ʿAfīfī 1939; Chodkiewicz 1993a, 1993b; Ghurāb 1985; Gril 2005; Lala 2019; Lipton 2018; Mayer 2008). On the topic of the influence that Ibn ʿArabī had, see (Knysh 1999; Morris 1986, 1987).

4　For details on these names, see (Al-Ghazālī 1999).

5　It is noteworthy that the concept of unity and multiplicity in essence and form in Islamic intellectual history was first thoroughly interrogated by Abū Yūsuf al-Kindī (d. 259/873?) (Al-Ahiwānī 1948, pp. 105–7).

6　It is important to note that Ibn ʿArabī does not suggest that God gains knowledge or is dependent on empirically-derived data in the sense that humans gain knowledge or are dependent on causes, as the citation makes plain.

7　The concept of the Perfect Man is one of the cornerstones of Ibn ʿArabī's mystical outlook. Masataka Takeshita explores this idea in (Takeshita 1987). However, it was ʿAbd al-Karīm al-Jīlī (d. 812/1408?) who really elaborated on and systematised the concept in his seminal work on this topic, *Al-Insān al-kāmil fī maʿrifat al-awākhir waʾl-awāʾil* (Al-Jīlī 1997). Fitzroy Morrissey interrogates al-Jīlī's understanding of the Perfect Man, and how it departs from Ibn ʿArabī's articulation of it, in (Morrissey 2020a).

8　A spiritual station (*maqām*) is a rank on the aspirant's spiritual journey. It is distinct from a spiritual state (*ḥāl*) because a spiritual state is temporary whereas a station is permanent. The aspirant must attain numerous stations on their spiritual journey (Al-Tahānawī n.d., 3:1227; Al-Qāshānī 1992, pp. 107–8).

9　This refers to the Prophet Muḥammad throwing some dust and small stones in the direction of enemy combatants at the Battle of Badr, which caused widespread panic alarm among them, as detailed by Abū Jaʿfar al-Ṭabarī (d. 310/923) in his celebrated commentary (Al-Ṭabarī 2000, 12:442–43).

[10]    This refers to the Prophet ʿĪsā's ability to bring forth the dead (Al-Ṭabarī 2000, 11:215).

[11]    See, for instance, Ibn ʿArabī (2002, p. 158).

[12]    There are five realms or levels of existence in Ibn ʿArabī's ontology. The first is the presence of the divine Essence (*dhāt*). This is followed by the presence of the spirits, then the presence of the souls, the presence of the 'images' (*mithāl*), and, finally, the presence of the sensible world (Chittick 1982, pp. 107–28).

[13]    Abū Bakr al-Bāqillānī (d. 403/1013) first introduced the idea of perpetual re-creation of all things in existence into Sunni theology in order to preserve absolute divine omnipotence (Gardet and Anawati 1981, pp. 62–64).

[14]    It is for this reason that in Sufi literature, the term for 'God' is usually 'the Truth' (al-Ḥaqq) (Jurjānī 1845, p. 96).

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
