# Peer review of "Turning Religious Experience into Reality: The Spiritual Power of Himma"

_religions, doi:10.3390/rel14030385_

Round 1
Reviewer 1 Report
This is a useful analysis of Ibn 'Arabī's conception of himma. I would like further clarity, however, on how the discussion of Ibn 'Arabī's understanding of himma relates to Schleiermacher's reflections - and the modern debate more generally - on religious experience. While the attempt to connect Ibn 'Arabī to modern views on the nature of religion is intriguing, the precise nature of Ibn 'Arabī's contribution to this debate is not particularly clear. Furthermore, I would also like to see some discussion of how Ibn 'Arabī's conceptualization of himma relates to the use of the concept in earlier Sufi thought - for instance with reference to the classical Sufi manuals. Finally, I would like to get a sense of whether Ibn 'Arabī's followers, such as Qūnawī and Jandī, use the concept in different ways to him.
Reviewer 2 Report
This is an excellent paper. It is very well-researched and clearly written. My only objection is that the introduction deals with a very broad topic on the nature of religious experience without really connecting with the main theme of himma. I feel that the paper should either go directly into the topic of himma, perhaps giving us a background of how others in the tradition have conceived it, etc. or draw out the relation how religious experience turns into realty through himma before focusing on Ibn al-Ê¿Arabi's discussion. Otherwise, very productive analysis of the main topic.
